# Calciprotein Particles Cause Physiologically Significant Pro-Inflammatory Response in Endothelial Cells and Systemic Circulation

**DOI:** 10.3390/ijms232314941

**Published:** 2022-11-29

**Authors:** Daria Shishkova, Arseniy Lobov, Bozhana Zainullina, Vera Matveeva, Victoria Markova, Anna Sinitskaya, Elena Velikanova, Maxim Sinitsky, Anastasia Kanonykina, Yulia Dyleva, Anton Kutikhin

**Affiliations:** 1Department of Experimental Medicine, Research Institute for Complex Issues of Cardiovascular Diseases, 6 Sosnovy Boulevard, 650002 Kemerovo, Russia; 2Laboratory of Regenerative Biomedicine, Institute of Cytology of the RAS, 4 Tikhoretskiy Prospekt, 194064 St. Petersburg, Russia; 3Centre for Molecular and Cell Technologies, St. Petersburg State University, Universitetskaya Embankment, 7/9, 199034 St. Petersburg, Russia

**Keywords:** calciprotein particles, calcium stress, endothelial cells, proteomic profiling, mitochondria, lysosomes, pro-inflammatory cytokines, blood cells, monocytes, systemic inflammatory response

## Abstract

Calciprotein particles (CPPs) represent an inherent mineral buffering system responsible for the scavenging of excessive Ca^2+^ and PO_4_^3−^ ions in order to prevent extraskeletal calcification, although contributing to the development of endothelial dysfunction during the circulation in the bloodstream. Here, we performed label-free proteomic profiling to identify the functional consequences of CPP internalisation by endothelial cells (ECs) and found molecular signatures of significant disturbances in mitochondrial and lysosomal physiology, including oxidative stress, vacuolar acidification, accelerated proteolysis, Ca^2+^ cytosolic elevation, and mitochondrial outer membrane permeabilisation. Incubation of intact ECs with conditioned medium from CPP-treated ECs caused their pro-inflammatory activation manifested by vascular cell adhesion molecule 1 (VCAM1) and intercellular adhesion molecule 1 (ICAM1) upregulation and elevated release of interleukin (IL)-6, IL-8, and monocyte chemoattractant protein-1/ C-C motif ligand 2 (MCP-1/CCL2). Among the blood cells, monocytes were exclusively responsible for CPP internalisation. As compared to the co-incubation of donor blood with CPPs in the flow culture system, intravenous administration of CPPs to Wistar rats caused a considerably higher production of chemokines, indicating the major role of monocytes in CPP-triggered inflammation. Upregulation of sICAM-1 and IL-8 also suggested a notable contribution of endothelial dysfunction to systemic inflammatory response after CPP injections. Collectively, our results demonstrate the pathophysiological significance of CPPs and highlight the need for the development of anti-CPP therapies.

## 1. Introduction

Calciprotein particles (CPPs) represent self-assembling scavengers of excessive Ca^2+^ and PO_4_^3−^ ions, which are formed in the human serum under the guidance of acidic proteins termed mineral chaperones (e.g., fetuin-A and albumin) [1,2,3,4,5]. While protecting the human organism from extraskeletal calcification, circulating CPPs are internalised by Kupffer cells [6,7,8] and endothelial cells (ECs) [7,8,9,10,11] and induce calcium stress upon their dissolution in lysosomes, thereby causing a pro-inflammatory response through the inflammasome-dependent and inflammasome-independent mechanisms [7,8,9,10,11,12,13,14]. In patients with hyperphosphatemia (e.g., those with end-stage renal disease), a considerable proportion of primary amorphous CPPs transform into secondary crystallised CPPs, which instigate stronger cytokine release and indicate critical depletion of mineral buffering systems [15,16]. A number of reports documented that the serum of patients with chronic kidney disease [17,18], arterial hypertension [18,19], coronary artery disease [11], cerebrovascular disease [11], and autoimmune disorders [20,21] shows increased CPP generation that reflects uncurbed mineral stress in these clinical scenarios and emphasizes the need in the development of anti-CPP therapies. For the experiments, CPPs are routinely synthesised in vitro by the supersaturation of serum-supplemented cell culture medium with Ca^2+^ and PO_4_^3−^ ions [8,9], as artificially generated CPPs are similar to those isolated from calcified human tissues [9]. Approximate concentration of CPPs in the human blood is 250 particles per µL (2.5 × 10^8^ per L), as measured by fluorescent-labelled bisphosphonate OsteoSense 680EX (IVISense Osteo 680 fluorescent probe). The kinetics of CPP assembling and clearance is far from being fully understood, albeit it evidently depends on the amount of Ca^2+^ and PO_4_^3−^ ions and acidic serum proteins (especially fetuin-A and albumin), and several reports specified a role of Mg^2+^ ions [22,23,24], HCO_3_^−^ ions [24,25], and pyrophosphate [25,26] in these processes as intrinsic Ca^2+^ antagonists.

Although previous studies delineated cytokines overproduced by ECs under CPP treatment (i.e., interleukin-6, interleukin-8, and monocyte chemoattractant protein 1) [9,10,11,12] and itemised other pathological consequences of CPP internalisation (i.e., hampered NO biosynthesis, endothelial-to-mesenchymal transition, and impaired mechanotransduction) [10,11], a little is known about the proteomic signatures of ECs to calcium stress and whether CPP-mediated cytokine elevation is itself able to elicit pathological effects in the intact ECs. Further, the impact of blood cells, in particular leukocyte populations, into the systemic inflammation after encountering circulating CPPs remains obscure.

Here, we for the first time performed an unbiased proteomic profiling of ECs treated with CPPs and revealed relative enrichment of the protein categories related to mitochondrial and lysosomal physiology after the CPP treatment. Conditioned cytokine-enriched medium from CPP-treated ECs promoted release of pro-inflammatory cytokines, increased production of cell adhesion molecules, and triggered apoptosis of intact ECs. In the flow culture system, circulating CPPs were internalised exclusively by ECs and monocytes, suggesting the latter as the only blood cell population directly contributing to CPP-related inflammation. Intravenous administration of CPPs caused a systemic inflammatory response that was significantly higher in animals than in the donor blood in vitro, pointing out a major contribution of monocyte-derived cytokines to the CPP-triggered cytokine upregulation. Taken together, our results suggest a potential clinical relevance of pre-clinical and clinical trials of chemical compounds restoring the mineral buffering systems, e.g., proteinogenic amino acids replenishing depleted serum acidic proteins or donors of Mg^2+^ ions concurring with Ca^2+^ for PO_4_^3−^ binding, in the context of cardiovascular disease.

## 2. Results

To better investigate the pathological effects of calciprotein particles, most of the experiments included four groups: control saline solution, non-toxic magnesiprotein particles (MPPs), amorphous primary CPPs (CPP-P) and crystalline secondary CPPs (CPP-S). CPP-P and CPP-S were quantified employing fluorescent-labelled bisphosphonate OsteoSense 680EX in order to correspond experimental doses (0.4–0.6 × 10^5^ particles per mL) to physiological serum CPP levels (2.5 × 10^5^ particles per mL), as ≈15–25% increase in CPP amounts has earlier been documented in patients with end-stage renal disease [27]. In rats, the experimental CPP dose was 0.8 × 10^5^ particles per mL, corresponding to ≈8% increase in CPP amount since the physiological level of CPPs in rat serum is 4-fold higher than in humans (≈1.0 × 10^6^ particles per mL). In addition, the study design included two EC lines: human coronary artery endothelial cells (HCAEC) and human internal thoracic artery endothelial cells (HITAEC), as coronary artery is atheroprone and internal thoracic artery is atheroresistant [28,29]. 

### 2.1. Proteomic Profiling Reveals a Significant Response of ECs to CPP-P or CPP-S Treatment

We first performed a label-free proteomic profiling of HCAEC and HITAEC treated with CPP-P or CPP-S for 24 h. Regardless of the cell line, CPP-P- and CPP-S-treated ECs have been clusterised far from control ECs (Figure 1A). The number of differentially expressed proteins (i.e., those with logarithmic fold change ≥ 1 and false discovery rate-corrected *p* value ≤ 0.05) between CPP-P and CPP-S treated ECs did not differ considerably, although HCAEC demonstrated stronger response [886 (CPP-P vs. control phosphate-buffered saline (PBS)) and 836 (CPP-S vs. PBS) differentially expressed proteins in HCAEC; 430 (CPP-P vs. PBS) and 441 (CPP-S vs. PBS) differentially expressed proteins in HITAEC, respectively, Figure 1B]. Bioinformatic enrichment analysis revealed significantly higher expression of the proteins within the mitochondria (in particular mitochondrial envelope and mitochondrial inner/outer membranes), lysosomes (especially lysosomal membrane), and phagosomes in both CPP-P- and CPP-S-treated ECs (Figure 1C). Certain proportion of mitochondrial proteins, particularly in HCAEC, was also downregulated under CPP treatment, indicating complex impairment of mitochondrial homeostasis (Figure 1C).

Detailed analysis of Gene Ontology and Reactome terms found combined over- and underexpression of nitrogen metabolism pathways as well as overrepresentation of the signalling pathways mediating response to oxidative stress, hypoxia, and endoplasmic reticulum stress (Table 1). Proteins responsible for vacuolar acidification and regulation of pH, proteolysis, and autophagy were significantly upregulated (Table 1). In keeping with these findings, proteins accountable for Ca^2+^ ion binding, mitochondrial outer membrane permeabilisation, and response to elevated cytosolic Ca^2+^ were also consistently overexpressed (Table 1). Taken together, these results suggested multiple violations of the signal transduction in the ECs upon CPP treatment, highlighting three major components of the respective molecular response: (1) oxidative and endoplasmic reticulum stress; (2) vacuolar acidification, autophagy, proteolysis, and altered pH; (3) response to Ca^2+^ cytosolic elevation in conjunction with Ca^2+^ ion binding and mitochondrial outer membrane permeabilisation (Table 1).

### 2.2. Pro-Inflammatory Activation of ECs by CPP-P and CPP-S Causes Pathological Paracrine Effects on Intact ECs

Then, we asked whether CPP-induced changes in the proteomic profile are pathophysiologically significant. As pathological activation of the ECs is accompanied by an augmented release by pro-inflammatory cytokines, we focused on the paracrine effects of conditioned medium collected from HCAEC and HITAEC after 24 h of co-incubation with MPPs, CPP-P, or CPP-S. To identify the exact mechanism behind these functional consequences, we added either complete conditioned medium (centrifuged at 3000× *g* to sediment the cell debris), conditioned medium depleted of extracellular vesicles (EVs) by centrifugation at 200,000× *g*, or EVs isolated from the conditioned medium using the specific kit (exoEasy Maxi, Qiagen) for another 24 h to the intact HCAEC and HITAEC.

We first performed the gene expression profiling to measure the level of the transcripts of *VCAM1*, *ICAM1*, *SELE*, and *SELP* genes (encoding EC receptors for leukocytes), *IL6*, *CXCL8*, *CCL2*, *CXCL1*, and *MIF* genes (encoding major endothelial pro-inflammatory cytokines), *NOS3* gene (encoding endothelial nitric oxide synthase), *SNAI1*, *SNAI2*, *TWIST1*, *ZEB1*, *CDH5*, and *CDH2* genes (encoding transcription factors and markers of endothelial-to-mesenchymal transition), *HES1*, *HEY1*, *HEY2*, and *NOTCH1* genes (encoding transcription factors and a marker of arterial specification), and *NR2F2* gene (encoding transcription factor of venous differentiation). In addition to the normalisation of the target gene expression for the expression of the housekeeping gene (*PECAM1* encoding CD31 protein), we have also performed the sequential normalisation for the gene expression after the addition of control PBS and upon the incubation with non-toxic MPPs (3^−ΔΔCt^) for the objective measurement of the gene expression at CPP-P or CPP-S treatment.

Complete conditioned medium withdrawn from CPP-P or CPP-S treated cells caused a remarkable pro-inflammatory activation of HCAEC detected by an elevated expression of *VCAM1*, *ICAM1*, *SELE*, *SELP*, *IL6*, *CXCL8*, *CCL2*, *CXCL1*, and *MIF* genes (Figure 2). However, EV-depleted conditioned medium and EVs did not provoke such pathological effects, and the response of HITAEC to any kind of conditioned medium or EVs was inconsistent at the transcript level (Figure 2). Hence, we then carried out a protein profiling of the cell lysate and cell culture supernatant exclusively from HCAEC incubated with the complete or EV-depleted conditioned medium.

Increased level of the cleaved (i.e., activated) executioner caspase 3 in HCAEC treated with CPP-P or CPP-S for 24 h clearly indicated their apoptosis (Figure 3A), suggesting a substantial deregulation of endothelial homeostasis and elevated production of pro-inflammatory cytokines into the medium. In concert with this finding, vascular cell adhesion molecule 1 (VCAM1) and intercellular adhesion molecule (ICAM1) mediating binding of leukocytes to ECs, as well as cleaved caspase 3, were upregulated in HCAEC upon the incubation with complete or EV-depleted conditioned medium from CPP-P- or CPP-S-treated HCAEC, confirming its pro-inflammatory effects (Figure 3B).

Cytokine profiling of either complete or EV-depleted conditioned medium from HCAEC by means of dot blotting found a notable elevation of interleukin-8 (IL-8) upon 24-h treatment with CPP-P or CPP-S and increase of IL-6 and monocyte chemoattractant protein 1/C-C motif ligand 2 (MCP-1/CCL2) to the detectable level after the incubation of such conditioned medium with intact HCAEC for another 24 h (Figure 4). These results illustrated pro-inflammatory paracrine effects of the complete and EV-depleted (i.e., containing soluble factors, such as cytokines but free of EVs) conditioned medium from CPP-P- or CPP-S-treated HCAEC.

As dot blotting is a semi-quantitative screening technique with limited sensitivity, we have also conducted a quantitative enzyme-linked immunosorbent assay and verified the augmented release of IL-6, IL-8, and MCP-1/CCL2 by HCAEC upon either 24-h CPP-P/CPP-S treatment or incubation with complete/EV-depleted conditioned medium (Figure 5) but not EVs (Appendix A) from CPP-P/CPP-S-treated HCAEC for another 24 h.

Collectively, these findings suggest soluble factors (i.e., pro-inflammatory cytokines IL-6, IL-8, and MCP-1/CCL2) but not EVs as the molecular substrate behind the pathological paracrine effects of conditioned medium from CPP-P- or CPP-S-treated ECs.

### 2.3. Circulating CPPs Are Internalised by Monocytes and Trigger a Systemic Inflammatory Response

We further investigated whether the blood cells contribute to the abovementioned pro-inflammatory paracrine effects of activated ECs upon CPP internalisation. For this task, we used the flow culture system imitating the blood vessels (ibidi Pump System Quad, ibidi) where we co-incubated donor blood and fluorescein isothiocyanate (FITC)-labeled MPPs, CPP-P, and CPP-S for 1 h at physiological shear stress values (15 dyn/cm^2^ similar to the human descending aorta). Flow cytometry analysis by means of forward and side scatter cell clusterisation distinguished monocytes, granulocytes, and lymphocytes; among the leukocyte populations, monocytes internalised MPPs, CPP-P, and CPP-S exclusively and concurrently with the decrease of the particle count in plasma (Figure 6).

Then, we differentiated leukocyte populations using an immunophenotyping approach (i.e., combined staining with anti-CD45, anti-CD16, anti-CD14, and anti-CD3 antibodies to stain leukocytes (CD45^+^) and differentiate neutrophils (CD45^+^CD16^high^), eosinophils (CD45^+^CD16^low^), monocytes (CD45^+^CD14^+^), T cells (CD45^+^CD3^+^), B cells (CD45^+^CD3^−^CD14^−^CD16^−^), and NK cells (CD45^+^CD3^−^CD14^−^CD16^low^). Eosinophils and NK cells were further differentiated by side scattering. In keeping with the data obtained from forward and side scattering clusterisation, monocytes were the only leukocyte population that internalised the particles that also retained in the plasma in significant amounts, whereas all other cells were devoid of specific FITC signal (Figure 7).

To verify the flow cytometry results, we conducted Ficoll gradient centrifugation isolation of agranulocytes (i.e., monocytes and lymphocytes) and stained them for a pan-leukocyte marker CD45. Monocytes, discernible by lobulated or indented nuclei, internalised all particle types (i.e., MPPs, CPP-P, and CPP-S), in contrast to lymphocytes having large and round nuclei (Figure 8). ECs cultured in the flow system in the presence of the whole blood also internalised the particles, which were distributed into the lysosomes (Figure 8). Red blood cells and platelets did not display any signs of MPP/CPP internalisation (Figure 8). Therefore, we proposed monocytes and ECs as the only cell populations capable of internalising CPP-P or CPP-S circulating in the human blood, in addition to the liver and spleen residual macrophages that have been reported as being largely responsible for the CPP clearance [6,7,8].

Finally, we assessed whether co-incubation of the human blood with CPP-P or CPP-S in the flow system or injections of these particles into the rat circulation alter a complete blood count or trigger an inflammatory response. No significant differences have been revealed with regards to any of full blood count parameters at any of the time points (0.5, 2, 4, and 24 h of co-incubation with human blood) (Appendix A). However, dot blotting profiling revealed an elevated production of pro-inflammatory cytokines (macrophage inflammatory protein (MIP)-1α/1β, stromal cell-derived factor (SDF)-1α / C-X-C motif chemokine ligand (CXCL)12, soluble intercellular adhesion molecule (ICAM-1), and interleukin-8 (IL-8)) and anti-inflammatory cytokine interleukin-1 receptor antagonist (IL-1Ra) in the donor blood after 1 or 24 h of co-incubation with CPP-P, although such response was donor-dependent (Figure 9A). Moreover, rats showed considerably higher pro-inflammatory response, in particular 1 h post-injection; it was highlighted by an augmented production of MIP-1α, MIP-3α, cytokine-induced neutrophil chemoattractants (CINC) 1 and 3, and CXCL10 (Figure 9B). Notably, several anti-inflammatory cytokines were also upregulated (IL-1Ra, IL-10, and ciliary neurotrophic factor (CNTF, Figure 9B). Among these molecules, MIP-1α showed the highest expression in both blood donors and animals, whilst CINC-3 and IL-1Ra demonstrated pronounced expression specifically in rats (Figure 9A,B). To identify the exact source of these cytokines and chemokines in plasma, we sedimented EVs by ultracentrifugation and repeated the dot blotting profiling in EVs and EV-depleted plasma of susceptible donors and animals. In blood donors, MIP-1α was found exclusively in the EVs as well as complement component C5/C5a, which has been raised both at CPP-P and CPP-S exposure (Figure 9C). In animals, MIP-1α, IL-1Ra, and sICAM-1 were detected both in the EVs and EV-depleted plasma, whereas other differentially expressed cytokines were documented exclusively in EV-depleted plasma (Figure 9D). Notably, sICAM-1 was increased only in EV-depleted plasma but not in EVs (Figure 9D). Intriguingly, sICAM-1 and MIP-3α were upregulated in the EV-depleted plasma both after CPP-P and CPP-S administration, whilst all other indicated cytokines were overrepresented exclusively upon the injection of CPP-P (Figure 9D).

These results demonstrated that intravenous administration of CPP-P caused a systemic inflammatory response, although excessive cytokine production was at best mild in response to CPP-S and both CPP-P and CPP-S did not trigger the demise of the blood cells. Among the blood cell populations, monocytes were exclusively responsible for the CPP internalisation, in addition to the ECs and residual liver and spleen macrophages as was noted earlier [6,7,8]. Notably, CPP-triggered systemic inflammatory response profile suggested a major contribution of monocyte-derived chemokines (MIP-1α, MIP-3α, CINC-1, CINC-3, CXCL10) in addition to the cytokines derived from vascular ECs (sICAM-1 and IL-8).

## 3. Discussion

Elevation of serum calcium and phosphate to the values near to or higher than the upper reference limit is strongly associated with increased risk of major adverse cardiovascular events, chronic heart failure, and cardiovascular death [30,31,32,33,34,35,36,37,38]. Out of serum calcium fractions, ionised calcium (Ca^2+^) is particularly important because it is not bound with serum proteins and, circulating in a free form, directly correlates with high risk of coronary artery disease, myocardial infarction, chronic brain ischemia, and ischaemic stroke [11,39]. In addition, high phosphate levels are associated with microvascular dysfunction [40], which, in turn, is itself an important risk factor of cardiovascular events [41]. In accord, serum magnesium (Mg^2+^, a physiological Ca^2+^ antagonist) near to or lower than the lower reference limit also correlates with higher risk of cardiovascular events and cardiovascular death [35,42,43,44,45,46,47,48,49,50,51,52]. Further, recent epidemiological studies and meta-analyses demonstrated that the reduction of acidic serum calcium-binding proteins (termed as mineral chaperones) to the lower reference limit or below is associated with increased risk of myocardial infarction, ischaemic stroke, and cardiovascular death [53,54,55,56,57,58]. Administration of cinacalcet, an alternative ligand of calcium-sensing receptor decreasing the production of parathyroid hormone and thereby lowering serum calcium and phosphate, to the patients with chronic kidney disease reduced the risk of major adverse cardiovascular events [59]. In contrast, pharmacological interventions increasing serum calcium and vitamin D—a bioactive factor raising calcium absorbance in the gut and its release from the bone to the circulation—did not diminish the risk of myocardial infarction, ischaemic stroke, and cardiovascular death [60,61,62].

Besides calciotropic and phosphotropic hormones and their receptors, magnesium (Mg^2+^), bicarbonate (HCO_3_^−^), and pyrophosphate [63], serum calcium and phosphate are regulated via a number of mineral chaperones having distinct mechanisms of action [64]. These proteins hinder bone resorption, inhibit calcium phosphate crystallisation, and abate calcium and phosphate reabsorption by the kidneys [64]. For instance, osteoprotegerin represents a decoy for the membrane-bound and secreted forms of receptor activator of nuclear factor κB (RANK) ligand, precluding its binding to the corresponding receptor RANK and preventing differentiation of pre-osteoclasts to osteoclasts, activation of osteoclasts, and bone resorption [65,66,67]. Osteopontin inhibits calcium phosphate crystallisation and enhances its dissolution via acidification of the tissue microenvironment by the stimulation of carbonic anhydrase II (an enzyme converting CO_2_ to H_2_CO_3_) release by monocytes/macrophages [68,69,70]. Serum ionised calcium (Ca^2+^) is controlled by the circulating protein scavengers [64], such as the most abundant blood protein albumin (35–50 g/L out of 65–85 g/L, i.e., up to 60% total serum protein), which binds Ca^2+^ ions by multiple negative charged amino acids in its tertiary structure, and osteonectin also binding Ca^2+^ ions via specific negatively charged domains [1,71]. In tissues, binding of extracellular Ca^2+^ and nascent calcium phosphate is regulated through the matrix Gla protein (MGP) and Gla-rich protein (GRP) belonging to the family of vitamin K-dependent proteins and bearing 5 (MGP) and 15 (GRP) negatively charged residues of γ-carboxylated glutamate (Gla) [72,73,74,75,76,77,78].

The most potent calcium-binding protein, however, is fetuin-A, which binds Ca^2+^ ions and nascent calcium phosphate by its negatively charged β-sheet located in the amino-terminal cystatin-like D1 and forms calciprotein monomers, ≈10 nm aggregates trapping approximately one-half of the excessive serum Ca^2+^ and PO_4_^3−^ ions and ≈95% fetuin-A [5,8,79]. Another half of Ca^2+^ and PO_4_^3−^ ions and the remaining 5% fetuin-A are contained within the CPPs, 50 to 500 nm carbonate-hydroxyapatite particles adsorbing ambient serum proteins [1,2,3,4,5]. Both calciprotein monomers and CPPs act as physiological mineral buffering systems [8,80]; however, increased generation of CPPs and transition of amorphous CPP-P to crystalline CPP-S in the serum have been reported in patients with chronic kidney disease [17,18], cardiovascular disease [11,18,19], and autoimmune disorders [20,21], suggestive of mineral homeostasis disturbances in these clinical scenarios and demanding the development of specific therapeutic approaches. Our previous studies have shown that CPPs are internalised by ECs under flow and trigger calcium stress leading to their pro-inflammatory activation and even apoptosis [9,10,11,12]. Nevertheless, the molecular response to CPPs, the pathophysiological significance of CPP-induced EC activation, and the contribution of CPPs to systemic inflammatory response have been unclear to date. Further, it is unknown whether CPPs are internalised by the blood cells during their circulation.

Here, we have for the first time performed a proteomic profiling of CPP-treated HCAEC and HITAEC and found the molecular signatures of mitochondrial and lysosomal dysfunction informative of oxidative stress, vacuolar acidification, hastened proteolysis, Ca^2+^ cytosolic elevation, and mitochondrial outer membrane permeabilisation. Incubation of intact ECs with conditioned medium from CPP-treated ECs (i.e., medium devoid of CPPs but enriched with IL-6, IL-8, and MCP-1/CCL2) caused their pro-inflammatory activation manifested by VCAM1 and ICAM1 overexpression and boosted secretion of IL-6, IL-8, and MCP-1/CCL2. Out of numerous blood cell populations, monocytes were exclusively responsible for CPP internalisation. Compared to the co-incubation model which employed addition of CPPs to the donor blood in the flow culture system, intravenous administration of CPPs to Wistar rats induced significantly higher production of monocyte-derived chemokines (MIP-1α, MIP-3α, CINC-1, CINC-3, CXCL10), whilst an upregulation of sICAM-1 and IL-8 suggested a notable contribution of endothelial dysfunction to the CPP-triggered systemic inflammatory response. These results are in concert with each other as mitochondrial and lysosomal stress enhances inflammasome-related pathways that are in control of excessive production of abovementioned cytokines and chemokines [81,82,83], and endothelial- and monocyte-derived cytokines can potentiate further pathological activation of endothelial cells and monocytes [84,85,86].

Among the advantages of our study were physiological doses of added CPPs equivalent to ≈15–25% increase in CPP concentration (+0.6 × 10^5^ particles per mL in static culture and +0.4 × 10^5^ particles per mL in flow culture, while the concentration of CPPs in the human blood was ≈2.5 × 10^5^ particles per mL), which has been reported in patients with end-stage renal disease [27], and corresponding to ≈8% increase in CPP concentration (0.8 × 10^5^ particles per mL, whereas the concentration of CPPs in the rat blood was 1.0 × 10^6^ particles per mL). In the conditioned medium experiments, we reduced the CPP dose to 0.1 × 10^5^ particles per mL (equivalent to ≈4% increase in CPP concentration) in order to better model a mild pro-inflammatory response, whilst in proteomic profiling experiments we focused on the molecular response to calcium stress.

Further, we employed both CPP-P and CPP-S, which have been synthesised in vitro as such approach generate particles similar to those isolated from human tissues [9]. Albeit both CPP-P and CPP-S were internalised by the ECs and monocytes and induced similar proteomic signatures and cytokine response in vitro, only CPP-P caused a significant systemic inflammatory response upon the intravenous administration into rats that might indicate their easier dissolution and better availability of Ca^2+^ ions. Notably, CPP-P are particularly relevant for the physiology as they precede CPP-S during artificial synthesis and formation in vivo, and it remains arguable whether significant quantities of CPP-S are presented in the human serum even in patients with end-stage renal disease.

Systemic inflammatory response is mediated through the network of cytokines either enclosed into the EVs or circulating in the blood as soluble molecules. Here, we found that both pro-inflammatory cytokines secreted by the ECs (IL-6, IL-8, and MCP-1/CCL2) and those collectively released into the blood (sICAM-1, MIP-1α, MIP-3α, CINC-1, CINC-3, and CXCL10) are abundant in the EV-depleted plasma, although MIP-1α and sICAM-1 have also been found in the EVs. Likewise, the pathological effects of EV-depleted conditioned medium were similar to those of complete conditioned medium, whilst EVs did not evoke any pro-inflammatory response in the ECs. Taken together, these results suggest that CPP-triggered paracrine and systemic pro-inflammatory response is pathophysiologically significant—as it induces pro-inflammatory activation of intact ECs—and is delivered by the cytokines freely circulating in the blood but not loaded into the EVs. Intriguingly, a number of anti-inflammatory molecules (IL-1Ra, IL-10, and CNTF) were also overrepresented in the plasma upon intravenous administration of CPPs, indicating that CPP-induced systemic inflammatory response is regulated and counteracted.

In conclusion, we show that excessive (supraphysiological but clinically relevant) concentrations of CPPs cause physiologically significant pro-inflammatory response in endothelial cells and systemic circulation. Among the major contributors of CPP-induced systemic inflammatory response are ECs and monocytes, which become activated upon CPP internalisation through the mitochondrial and lysosomal dysfunction and produce excessive amounts of cytokines and chemokines. These results highlight a need in the development of anti-CPP therapies.

## 4. Materials and Methods

### 4.1. Artificial Synthesis and Quantification of CPPs

To synthesise primary (CPP-P) and secondary (CPP-S) CPPs, stock solutions of CaCl_2_ (21115, Sigma-Aldrich, St. Louis, MO, USA) and Na_2_HPO_4_ (94046, Sigma-Aldrich, St. Louis, MO, USA) were diluted to equal concentrations of 3 (CPP-P) or 7.5 (CPP-S) mmol/L in Dulbecco’s modified Eagle’s medium (DMEM, 31330038, Thermo Fisher Scientific, Waltham, MA, USA) supplemented with 10% (CPP-P) or 1% foetal bovine serum (CPP-S). For the synthesis of magnesiprotein particles (MPPs), stock solutions of MgCl_2_ (97062-848, VWR, Radnor, PA, USA) and Na_2_HPO_4_ were diluted to equal concentrations of 20 mmol/L in DMEM supplemented with 10% foetal bovine serum (FBS, 1.1.6.1, BioLot, St. Petersburg, Russia). The reagents were added into DMEM in the following order: (1) FBS; (2) CaCl_2_ or MgCl_2_; (3) Na_2_HPO_4_, with a vortexing between the added reagents. Following incubation for 24 h in cell culture conditions, the medium was centrifuged at 200,000× *g* for 1 h (Optima MAX-XP, Beckman Coulter, Brea, CA, USA), and the particle sediment was resuspended in the sterile phosphate-buffered saline (PBS, pH = 7.4, 2.1.1, BioLot, St. Petersburg, Russia).

Quantification of CPP-P, CPP-S, and MPPs was performed as in [11]. Briefly, the concentration of CPP-P/CPP-S/MPP was ≈1.2 × 10^3^ particles per µL suspension. In the experiments, we used the following doses of particles: (1) proteomic profiling experiments: 0.6 × 10^5^ particles per mL cell culture medium, 25 µg/mL calcium or 1:4 particle-to-cell ratio; (2) conditioned medium experiments: 0.1 × 10^5^ particles per mL cell culture medium, ≈4 µg/mL calcium, or 1:25 particle-to-cell ratio; (3) flow culture experiments: 0.4 × 10^5^ particles per mL blood or 16.5 µg/mL calcium); (4) animal experiments: 0.8 × 10^5^ particles per mL blood or 35 µg/mL calcium). As the concentration of CPPs in the human blood was ≈2.5 × 10^5^ particles per mL and our doses were ≈0.6 × 10^5^ particles per mL in static culture and 0.4 × 10^5^ particles per mL in flow culture, we attributed the applied doses to ≈15–25% increase in CPP concentration, which was reported in patients with end-stage renal disease [27]. In rats, the concentration of CPPs was ≈1.0 × 10^6^ particles per mL and our dose was 0.8 × 10^5^ particles per mL, equivalent to ≈8% increase in CPP amount. In the conditioned medium experiments, we reduced the particle/calcium dose to better model a mild pro-inflammatory response, whilst in proteomic profiling experiments we focused on the molecular response to calcium stress.

### 4.2. Cell Culture

Primary cultures of human coronary artery endothelial cells (HCAEC, 300K-05a, Cell Applications, San Diego, CA, USA) and human internal thoracic artery endothelial cells (HITAEC, 308K-05a, Cell Applications, San Diego, CA, USA) were grown in T-25, T-75, or T-150 flasks (90026, 90076, and 90552, respectively, Techno Plastic Products, Trasadingen, Switzerland), according to the manufacturer’s protocol using MesoEndo Growth Medium (212-500, Cell Applications, San Diego, CA, USA) and subculture reagent kit (090K, Cell Applications, San Diego, CA, USA). Immediately before all experiments, we replaced MesoEndo Growth Medium with MesoEndo Growth Medium without FBS (212F-500, Cell Applications, San Diego, CA, USA). During such replacement, we washed cells twice with warm (≈37 °C) PBS to remove the residual serum components, which could affect further proteomic profiling or contaminate serum-free medium with serum-derived EVs. The rationale behind the use of these two EC lines was that coronary artery is atheroprone and internal thoracic artery is atheroresistant [28,29].

### 4.3. Proteomic Profiling

For the proteomic profiling experiments, HCAEC and HITAEC were cultured in 6-well plates (92406, Techno Plastic Products, Trasadingen, Switzerland) to ≈90% confluence (≈0.5 × 10^6^ cells per well) and were then exposed to 100 µL of MPPs, CPP-P, or CPP-S (0.6 × 10^5^ particles per mL or 25 µg/µL calcium) or PBS (n = 3 wells per group) in a serum-free medium (212F-500, Cell Applications, San Diego, CA, USA) for 24 h. Then, cell cultures were washed with ice-cold (4 °C) PBS (pH = 7.4, 2.1.1, BioLot, St. Petersburg, Russia) and lysed in RIPA buffer (89901, Thermo Fisher Scientific, Waltham, MA, USA) supplemented with protease and phosphatase inhibitors (78444, Thermo Fisher Scientific, Waltham, MA, USA), according to the manufacturer’s protocol. Quantification of total protein was conducted using BCA Protein Assay Kit (23227, Thermo Fisher Scientific, Waltham, MA, USA) and Multiskan Sky microplate spectrophotometer (Thermo Fisher Scientific, Waltham, MA, USA) in accordance with the manufacturer’s protocol.

Upon the removal of RIPA buffer by acetone precipitation (650501, Sigma-Aldrich, St. Louis, MO, USA), protein pellet was resuspended in 8 mol/L urea (U5128, Sigma-Aldrich, St. Louis, MO, USA) diluted in 50 mmol/L ammonium bicarbonate (09830, Sigma-Aldrich, St. Louis, MO, USA). The protein concentration was measured by Qubit 4 fluorometer (Q33238, Thermo Fisher Scientific, Waltham, MA, USA) with QuDye Protein Quantification Kit (25102, Lumiprobe, Cockeysville, MD, USA), according to the manufacturer’s protocol. Protein samples (15 µg) were then incubated in 5 mmol/L dithiothreitol (D0632, Sigma-Aldrich, St. Louis, MO, USA) for 1 h at 37 °C with the subsequent incubation in 15 mmol/L iodoacetamide for 30 min in the dark at room temperature (I1149, Sigma-Aldrich, St. Louis, MO, USA). Next, the samples were diluted with 7 volumes of 50 mmol/L ammonium bicarbonate and incubated for 16 h at 37 °C with 200 ng of trypsin (1:50 trypsin:protein ratio; VA9000, Promega, Madison, WI, USA). The peptides were then frozen at −80 °C for 1 h and desalted with stage tips (Tips-RPS-M.T2.200.96, Affinisep, Le Houlme, France), according to the manufacturer’s protocol using methanol (1880092500, Sigma-Aldrich, St. Louis, MO, USA), acetonitrile (1000291000, Sigma-Aldrich, St. Louis, MO, USA), and 0.1% formic acid (33015, Sigma-Aldrich, St. Louis, MO, USA). Desalted peptides were dried in centrifuge concentrator (Concentrator plus, Eppendorf, Hamburg, Germany) for 3 h and finally dissolved in 20 µL 0.1% formic acid for further shotgun proteomics analysis.

Shotgun proteomics analysis was performed in duplicate by ultra-high performance liquid chromatography-tandem mass spectrometry (UHPLC-MS/MS) with ion mobility in TimsToF Pro mass spectrometer with nanoElute UHPLC system (Bruker Daltonics) using ≈500 ng of peptides. UHPLC was performed in the two-column separation mode with Acclaim PepMap 5 mm Trap Cartridge (Thermo Fisher Scientific) and Bruker Fifteen separation column (C18 ReproSil AQ, 150 mm × 0.75 mm, 1.9 µm, 120 A; Bruker Daltonics) in a gradient mode with 400 nL/min flow rate and 40 °C. Phase A was water/0.1% formic acid, phase B was acetonitrile/0.1% formic acid (1000291000, Sigma-Aldrich). The gradient was from 2% to 30% phase B for 42 min, then to 95% phase B for 6 min with subsequent washing with 95% phase B for 6 min. Before each sample, trap and separation columns were equilibrated with 10 and 4 column volumes, respectively. CaptiveSpray ion source was used for electrospray ionization with 1600 V of capillary voltage, 3 L/min N2 flow, and 180 °C source temperature. The mass spectrometry acquisition was performed in DDA-PASEF mode with 0.5 s cycle in positive polarity with the fragmentation of ions with at least two charges in m/z range from 100 to 1700 and ion mobility range from 0.85 to 1.30 1/K0.

Protein identification was performed in PEAKS Studio Xpro software (a license granted to St. Petersburg State University; Bioinformatics Solutions Inc., Waterloo, ON, Canada) using human protein SwissProt database (https://www.uniprot.org/; accessed on 20 July 2022; organism: Human [9606]; uploaded on 2 March 2021; 20,394 sequences) and protein contaminants database CRAP (version of 4 March 2019). The search parameters were: parent mass error tolerance 10 ppm and fragment mass error tolerance 0.05 ppm, protein and peptide FDR < 1% and 0.1% respectively, two possible missed cleavage sites, proteins with ≥2 unique peptides. Cysteine carbamidomethylation was set as fixed modification. Methionine oxidation, N-terminal acetylation, asparagine and glutamine deamidation were set as variable modifications.

The mass spectrometry proteomics data have been deposited to the ProteomeXchange Consortium via the PRIDE [87] partner repository with the dataset identifier PXD038017 and 10.6019/PXD038017. Label-free quantification by peak area under the curve and spectral counts was used for the further analysis in R (version 3.6.1; R Core Team, 2019). All proteins presented in all (3/3) biological replicates were identified and the groups were compared by “VennDiagram” package [88] and drawing of Venn diagram. The proteins with NA in ≥10% of samples were removed and imputation of missed values by k-nearest neighbours was performed by the “impute” package [89]. Then, log-transformation and quantile normalization with further analysis of differential expression by “limma” package [90] were conducted. Finally, we carried out clusterisation of samples by sparse partial least squares discriminant analysis in the “MixOmics” package [91]. “ggplot2” [92] and “EnhancedVolcano” [93] packages were used for visualization. Reproducible code for data analysis is available from https://github.com/ArseniyLobov/hcaec-and-hitaec-treated-with-calciprotein-particles (accessed on 18 October 2022). Differentially expressed proteins were defined as those with logarithmic fold change ≥1 and false discovery rate-corrected *p* value ≤ 0.05. Bioinformatic analysis was performed using Gene Ontology [94,95], Reactome [96,97], UniProtKB Keywords [98], and Kyoto encyclopaedia of genes and genomes (KEGG) databases [99,100].

### 4.4. Modelling of Endothelial Cell Paracrine Effects

To model the paracrine effects of the conditioned medium collected from CPP-treated ECs, HCAEC and HITAEC were cultured in T-150 flasks (90552, Techno Plastic Products, Trasadingen, Switzerland) to ≈90% confluence (≈7.5 × 10^6^ cells) and were then exposed to 250 µL of MPPs, CPP-P, or CPP-S (0.1 × 10^5^ particles per mL, 4 µg/µL calcium, or 1:25 particle-to-cell ratio) or PBS in a serum-free medium (212F-500, Cell Applications, San Diego, CA, USA) for 24 h. Then, conditioned medium was withdrawn from the flasks and either: (1) centrifuged at 3000× *g* to sediment the large debris and obtain a complete conditioned medium; (2) centrifuged at 200,000× *g* for 2 h to deplete conditioned medium from the EVs; (3) passed through the EV purification columns (exoEasy Maxi Kit, 76064, Qiagen, Hilden, Germany), according to the manufacturer’s protocol with the subsequent centrifugation of the eluate at 200,000× *g* for 2 h to sediment EVs. Complete and EV-depleted conditioned medium as well as EVs from MPP-, CPP-P, CPP-S, or PBS-treated ECs were then added to the intact EC cultures (≈90% confluence, ≈1.0 × 10^6^ cells per flask) from the same donors grown in T-25 flasks (90026, Techno Plastic Products) for another 24 h.

### 4.5. Molecular Profiling

After exposure of intact HCAEC and HITAEC cultures to either complete or EV-depleted conditioned medium or EVs from MPP-, CPP-P, CPP-S, or PBS-treated ECs (n = 6 flasks per group) for 24 h, conditioned medium was withdrawn. Cell cultures were washed with ice-cold (4 °C) PBS (pH = 7.4, 2.1.1, BioLot, St. Petersburg, Russia) and lysed with either TRIzol (15596018, Thermo Fisher Scientific, Waltham, MA, USA) to isolate RNA (n = 3 flasks per group) or RIPA buffer (89901, Thermo Fisher Scientific, Waltham, MA, USA) supplemented with protease and phosphatase inhibitors (78444, Thermo Fisher Scientific, Waltham, MA, USA) to extract total protein (n = 3 flasks per group), according to the respective manufacturers’ protocols.

To measure gene expression, we quantified total RNA using Qubit 4 fluorometer (Thermo Fisher Scientific, Waltham, MA, USA) supplied with Qubit RNA BR and IQ assay kits (Q10210 and Q33222 respectively, Thermo Fisher Scientific, Waltham, MA, USA), performed a reverse transcription using a High Capacity cDNA Reverse Transcription Kit (4368814, Thermo Fisher Scientific, Waltham, MA, USA), and conducted quantitative polymerase chain reaction (qPCR) using customised primers (500 nmol/L each, Evrogen, Moscow, Russia, Appendix A), cDNA (20 ng), and PowerUp SYBR Green Master Mix (A25778, Thermo Fisher Scientific, Waltham, MA, USA) according to the manufacturers’ protocols as in [11]. Technical replicates (n = 3 per sample collected from one flask) were performed in all qPCR experiments. Average ΔCt values and their standard deviations are presented in the Appendix A.

For measuring protein expression, quantification of total protein was conducted as described above and protein loading, separation, and transfer were performed as in [11]. Blots were probed with rabbit antibodies to vascular cell adhesion molecule 1 (1:1000, VCAM1, ab134047, Abcam, Cambridge, UK) and intercellular cell adhesion molecule 1 (1:1000, ICAM1, ab109361, Abcam, Cambridge, UK) or mouse antibodies to total and cleaved caspase-3 (1:200, ab208161, Abcam, Cambridge, UK), CD31 (loading control, 1:1000, ab9498, Abcam, Cambridge, UK), and glyceraldehyde-3-phosphate dehydrogenase (GAPDH, loading control, 1:250, ab139416, Abcam, Cambridge, UK). Horseradish-peroxidase-conjugated goat anti-rabbit (7074, Cell Signaling Technology, Danvers, MA, USA) or goat anti-mouse secondary antibodies (AP130P, Sigma-Aldrich, St. Louis, MO, USA) were used at 1:200 and 1:1000 dilution, respectively. Incubation with the antibodies and chemiluminescent detection were carried out as in [11].

Conditioned medium was profiled for 36 cytokines as well as 55 pro- and anti-angiogenic molecules using the respective dot blotting kits (ARY005B and ARY007 respectively, R&D Systems, Minneapolis, MN, USA), according to the manufacturer’s protocols. Chemiluminescent detection was performed using Odyssey XF imaging system (LI-COR Biosciences, Lincoln, NE, USA). Levels of pro-inflammatory cytokines (interleukin-6, interleukin-8, and MCP-1/CCL2) were measured using the respective kits for enzyme-linked immunosorbent assay (430507, 431507, and 438807, BioLegend, San Diego, CA, USA), according to the manufacturer’s protocols (n = 6 measurements per group) and colorimetric analysis was conducted using Multiskan Sky microplate spectrophotometer (Thermo Fisher Scientific, Waltham, MA, USA).

### 4.6. Internalisation Assays

For the detection of MPPs, CPP-P, and CPP-S after the internalisation, we labeled the particles with a fluorescein isothiocyanate (FITC) by incubation of the particle sediment with 25 µL (125 µg) FITC-labeled albumin (5 mg/mL, A23015, Thermo Fisher Scientific, Waltham, MA, USA) for 1 h. Particles were then washed once in PBS to remove unconjugated albumin.

The study was conducted according to the latest revision of Declaration of Helsinki (2013), and the donor study protocol was approved by the Local Ethical Committee of the Research Institute for Complex Issues of Cardiovascular Diseases (Kemerovo, Russian, protocol code 2022/08, date of approval: 21 April 2022). A written informed consent was provided by all study participants after receiving a full explanation of the study purposes.

To assess internalisation of MPPs, CPP-P, and CPP-S under flow, we collected the blood of two healthy volunteers into K2-EDTA tubes (363706, Becton Dickinson, Franklin Lakes, NJ, USA), diluted it with physiological saline (0.9% NaCl, Hematek, Tver, Russia) in a 1:1 proportion to reduce the viscosity, and co-incubated it with 500 µL of FITC-labeled MPPs, CPP-P, or CPP-S (0.4 × 10^5^ particles per mL blood or 16.5 µg/µL calcium) or physiological saline for 1 h at 15 dyn/cm^2^ shear stress (similar to the values registered in the human descending aorta) using the ibidi Pump System Quad flow culture system (Ibidi, Grafelfing, Germany). HCAEC were seeded into the channel slides (µ-Slide I Luer, 80176, Ibidi, Grafelfing, Germany) to ≈100% confluence (≈0.1 × 10^6^ cells) 24 h before the experiment to ensure a proper adhesion to the polymer coverslip. Channel slides with HCAEC were connected to the flow culture system immediately before particle addition. After 1 h of incubation with the particles, blood was withdrawn and cell populations were separated using a double density gradient (Histopaque-1077 and Histopaque-1119, density: 1.077 and 1.119 g/mL, 10771 and 11191, respectively, Sigma-Aldrich, St. Louis, MO, USA), according to the manufacturer’s protocol. Platelets were isolated from the separate blood aliquot by sequential centrifugation at 100× *g* and 400× *g* during 10 min.

Agranulocytes (monocytes and lymphocytes), red blood cells, and platelets were seeded into the 8-well cell culture chambers (µ-Slide 8 Well, 80826, Ibidi, Grafelfing, Germany) for 3 h to ensure a proper adhesion to the polymer coverslip and fixed with 4% paraformaldehyde (158127, Sigma-Aldrich, St. Louis, MO, USA) for 10 min, permeabilised in Triton X-100 (T8787, Sigma-Aldrich, St. Louis, MO, USA) for 15 min, and blocked in 1% bovine serum albumin (P091E, PanEco, Moscow, Russia) for 1 h to prevent non-specific binding. Then, blood cells were stained with rabbit anti-CD45 (1:200, ab10558, Abcam, Cambridge, UK; 1:200, SL4819R, Sunlong Biotech, Hangzhou, Zhejiang, China), rabbit or mouse anti-CD235a (1:200, ab129024, Abcam, Cambridge, UK; 1:50, sc-53295, Santa Cruz Biotechnology, Dallas, TX, USA), or rabbit anti-CD41 (1:200, ab134131, Abcam, Cambridge, UK; 1:200, SL2636R, Sunlong Biotech, Hangzhou, Zhejiang, China) primary antibodies overnight at 4 °C. The next day, cells were stained with donkey anti-rabbit pre-adsorbed Alexa Fluor 555-conjugated (1:500, ab150062, Abcam, Cambridge, UK) secondary antibodies for 1 h at room temperature. Counterstaining was performed with 4′,6-diamidino-2-phenylindole (DAPI, 10 µg/mL, D9542, Sigma-Aldrich, St. Louis, MO, USA) for 30 min. At all stages, washing was conducted with PBS (pH 7.4, 60201, Pushchino Laboratories, Pushchino, Russia). Coverslips were mounted with ProLong Gold Antifade (P36934, Thermo Fisher Scientific, Waltham, MA, USA). HCAEC in the channel slides (µ-Slide I Luer, 80176, Ibidi, Grafelfing, Germany) were stained with a pH sensor LysoTracker Red (500 nmol/L, L7528, Thermo Fisher Scientific, Waltham, MA, USA) for 1 h, according to the manufacturer’s protocol. Visualisation was performed by confocal microscopy (LSM 700, Carl Zeiss, Oberkochen, Germany).

Alternatively, immediately after adding FITC-labeled MPPs, CPP-P, or CPP-S or control physiological saline solution to the blood and after 1 h of co-incubation with the particles under flow as described above, we separated plasma by blood centrifugation at 3000× *g* for 15 min (5804R, Eppendorf, Hamburg, Germany) and white blood cells by lysing red blood cells (A09777, Beckman Coulter, Brea, CA, USA), according to the manufacturer’s protocol, and measured FITC fluorescence in combination with forward scatter (FSC) and side scatter (SSC) by flow cytometry using CytExpert software (CytoFLEX, Beckman Coulter, Brea, CA, USA). A portion of white blood cells was additionally stained with the cocktail of anti-CD45 (mouse anti-human CD45 Pacific Blue, 304029, BioLegend, San Diego, CA, USA), anti-CD16 (mouse anti-human CD16 APC, B00845, Beckman Coulter, Brea, CA, USA), anti-CD14 (mouse anti-human CD14 APC/Cyanine7 (301819, BioLegend, San Diego, CA, USA), and anti-CD3 (mouse anti-human CD3 Alexa Fluor 700, 344844, BioLegend, San Diego, CA, USA) antibodies to stain leukocytes (CD45^+^) and to differentiate neutrophils (CD45^+^CD16^high^), eosinophils (CD45^+^CD16^low^), monocytes (CD45^+^CD14^+^), T cells (CD45^+^CD3^+^), B cells (CD45^+^CD3^−^CD14^−^CD16^−^), and NK cells (CD45^+^CD3^−^CD14^−^CD16^low^). For the identification of blood cell populations internalising MPPs, CPP-P, and CPP-S, Eosinophils and NK cells were further differentiated by side scattering. Isotype controls for the abovementioned mouse anti-human CD45, mouse anti-human CD16, mouse anti-human CD14, and mouse anti-human CD3 antibodies were mouse IgG1 Pacific Blue (400151, BioLegend, San Diego, CA, USA), mouse IgG1 APC (IM24754, Beckman Coulter, Brea, CA, USA), mouse IgG2a APC/Cyanine7 (400229, BioLegend, San Diego, CA, USA), and mouse IgG1 Alexa Fluor 700 (400143, BioLegend, San Diego, CA, USA) antibodies, respectively.

For the immunophenotyping, 200 µL of blood was incubated with the antibodies (10 µL antibodies per 100 µL blood for B00845 and IM24754 (Beckman Coulter) and 5 µL antibodies per 100 µL blood for 304029, 301819, 344844, 400151, 400229, and 400143 (BioLegend)) at 4 °C for 30 min. Red blood cells were lysed by adding 2 mL VersaLyse lysing solution (A09777, Beckman Coulter, Brea, CA, USA) into each sample for 10 min. Then, samples were centrifuged at 300× *g* for 5 min at 4 °C, and pellet was resuspended in 200 µL PBS (pH = 7.4, 2.1.1, BioLot, St. Petersburg, Russia). Gating strategy included drawing of the SSC/CD45 signal intensity histogram with the subsequent depiction of SSC/CD16 and CD14/CD16 histograms. To obtain the CD16/CD3 histogram, we used both SSC/CD45 signal intensity gating and FSC/SSC subgating. During the flow cytometry, we measured 20,000 events in each white blood cell sample and 100,000 events in each plasma sample.

### 4.7. Cytotoxicity and Inflammation Assays

Toxicity of MPPs, CPP-P, and CPP-S for the blood cells and their ability to trigger systemic inflammatory response were assessed using the complete blood count measurements and dot blotting profiling of cytokines. To evaluate the changes in complete blood count, we collected the blood of six healthy volunteers into K2-EDTA tubes (363706, Becton Dickinson, Franklin Lakes, NJ, USA), diluted it with physiological saline (0.9% NaCl, Hematek, Tver, Russia) in a 1:2 proportion to reduce the viscosity, and co-incubated it with 500 µL of FITC-labeled MPPs, CPP-P, or CPP-S (0.4 × 10^5^ particles per mL blood, 16.5 µg/µL calcium) or physiological saline for 24 h at 15 dyn/cm^2^ shear stress using the ibidi Pump System Quad flow culture system (Ibidi, Grafelfing, Germany). At time points of 30 min, 2, 4, and 24 h, blood aliquots were collected into K2-EDTA tubes (363706, Becton Dickinson, Franklin Lakes, NJ, USA) and the complete blood count was measured using an automated hematology analyzer (Sysmex XN-550, Kobe, Japan). 

To assess whether MPPs, CPP-P, and CPP-S induce systemic inflammatory response in vitro, we collected the blood of two healthy volunteers into K2-EDTA tubes (363706, Becton Dickinson, Franklin Lakes, NJ, USA), diluted it with physiological saline (0.9% NaCl, Hematek, Tver, Russia) in a 1:1 proportion to reduce the viscosity, and co-incubated it with 500 µL of FITC-labeled MPPs, CPP-P, or CPP-S (0.4 × 10^5^ particles per mL blood, 16.5 µg/µL calcium) or physiological saline for 1 or 24 h at 15 dyn/cm^2^ shear stress using the ibidi Pump System Quad flow culture system (Ibidi, Grafelfing, Germany). In vivo, we injected MPPs, CPP-P, or CPP-S (0.8 × 10^5^ particles per mL blood, 35 µg/mL calcium) or physiological saline into the tail vein of two 3-month female Wistar rats (200 g body weight, 13 mL circulating blood volume) for 1 or 2 h and then collected the blood into the centrifuge tubes (91015, Techno Plastic Products, Trasadingen, Switzerland). Serum was obtained by the blood centrifugation at 3000× *g* for 15 min (5804R, Eppendorf, Hamburg, Germany), aliquoted and then depleted from EVs by sequential centrifugation at 15,000× *g* for 15 min (Microfuge 20R, Beckman Coulter, Brea, CA, USA) and 200,000× *g* for 1.5 h (Optima MAX-XP, Beckman Coulter, Brea, CA, USA). EVs sedimented at the latter two centrifugation steps were lysed in RIPA buffer (89901, Thermo Fisher Scientific, Waltham, MA, USA) supplemented with protease and phosphatase inhibitors (78444, Thermo Fisher Scientific, Waltham, MA, USA), according to the manufacturer’s protocol to isolate the total protein. Whole and EV-depleted plasma as well as EV protein lysate were further profiled for 36 cytokines (human plasma) or 29 cytokines (rat plasma) using the respective dot blotting kits (ARY005B and ARY008, respectively, R&D Systems, Minneapolis, MN, USA), according to the manufacturer’s protocols. Chemiluminescent detection was performed using Odyssey XF imaging system (LI-COR Biosciences, Lincoln, NE, USA). The animal study protocol was approved by the Local Ethical Committee of the Research Institute for Complex Issues of Cardiovascular Diseases (Kemerovo, Russian Federation, protocol code 2022/09, date of approval: 21 April 2022). Animal experiments were performed in accordance with the European Convention for the Protection of Vertebrate Animals (Strasbourg, 1986) and Directive 2010/63/EU of the European Parliament on the protection of animals used for scientific purposes.

### 4.8. Statistical Analysis

Statistical analysis was performed using GraphPad Prism 8 (GraphPad Software, San Diego, CA, USA). For descriptive statistics, data are presented as median, 25th and 75th percentiles, and range. Four independent groups were compared by the Kruskal–Wallis test with post hoc calculation of false discovery rate (FDR) by the two-stage linear step-up procedure of Benjamini, Krieger, and Yekutieli. *p* values, or *q* values if FDR was applied (*q*-values are the name given to the adjusted *p* values found using an optimised FDR approach), ≤0.05 were regarded as statistically significant.

## Figures and Tables

**Figure 1 ijms-23-14941-f001:**
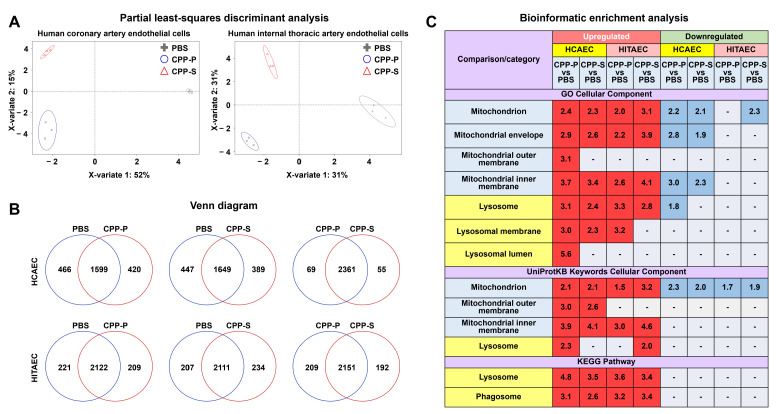
Proteomic profiling of HCAEC and HITAEC treated with CPP-P, CPP-S, or PBS for 24 h. (**A**) Partial least-squares discriminant analysis shows differential clustering of HCAEC (left) and HITAEC (right) treated with CPP-P (blue circles), CPP-S (red triangles), or PBS (gray crosses); (**B**) Venn diagrams demonstrating the numbers of differentially expressed proteins in each group in pairwise comparisons for HCAEC (top) and HITAEC (bottom); (**C**) Bioinformatic enrichment analysis (Gene Ontology, UniProtKB Keywords, and KEGG databases).

**Figure 2 ijms-23-14941-f002:**
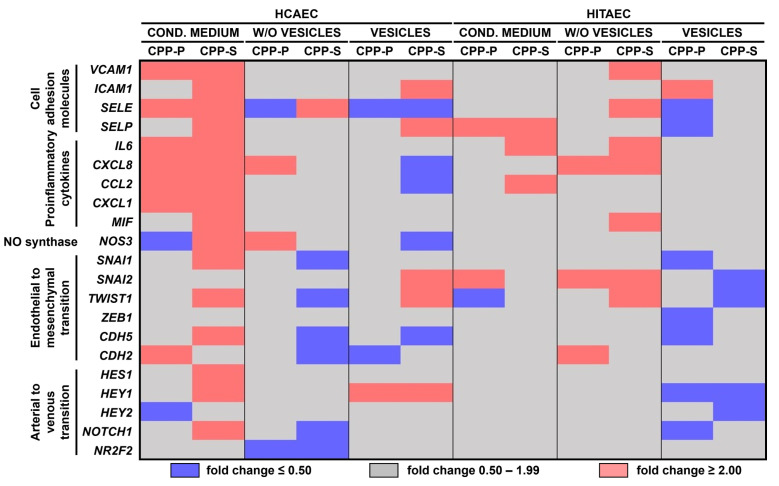
Gene expression profiling of HCAEC (left) and HITAEC (right) incubated with either complete conditioned medium, EV-depleted conditioned medium (w/o vesicles) or purified EVs (vesicles) from CPP-P- or CPP-S-treated HCAEC and HITAEC for 24 h. The duration of CPP pre-treatment was also 24 h. Reverse transcription polymerase chain reaction. Heat map. Violet, gray, and pink colours mean fold change ≤0.50, 0.50 to 1.99, and ≥2.00, respectively.

**Figure 3 ijms-23-14941-f003:**
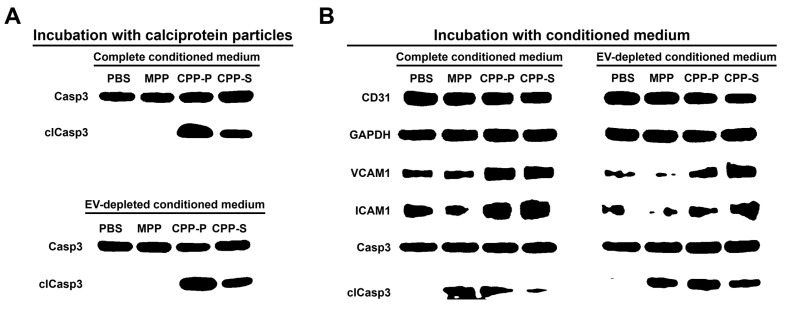
Western blotting profiling of (**A**) HCAEC incubated with MPPs, CPP-P, CPP-S, or control PBS solution for 24 h, experiments with complete conditioned medium (top) and EV-depleted conditioned medium (bottom); (**B**) HCAEC incubated with either complete or EV-depleted conditioned medium from HCAEC treated with MPPs, CPP-P, CPP-S, or control PBS solution for 24 h. CD31 and GAPDH (glyceraldehyde 3-phosphate dehydrogenase) are the loading control. Total caspase 3 (Casp3) is another loading control specifically in relation to the measurement of cleaved caspase 3.

**Figure 4 ijms-23-14941-f004:**
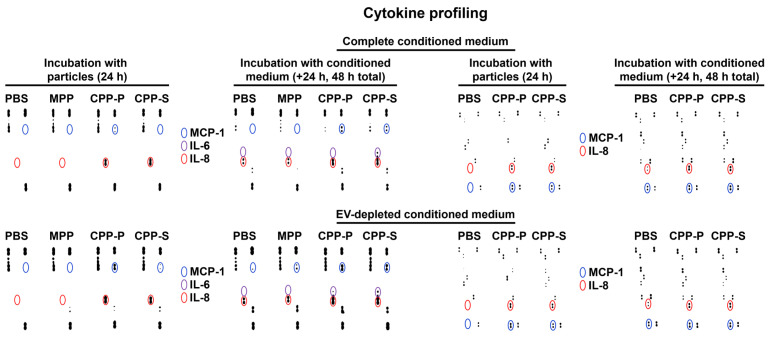
Cytokine profiling of either complete (top) or EV-depleted (bottom) conditioned medium from HCAEC incubated with MPPs, CPP-P, CPP-S, or control PBS solution for 24 h and upon another 24 h of incubation (48 h total) with intact HCAEC. Specific dot blotting kits for the measurement of cytokines (left) and pro- or anti-angiogenic molecules (right). Blue, violet, and red circles demarcate the signal from the antibodies to MCP-1/CCL2, IL-6, and IL-8, respectively.

**Figure 5 ijms-23-14941-f005:**
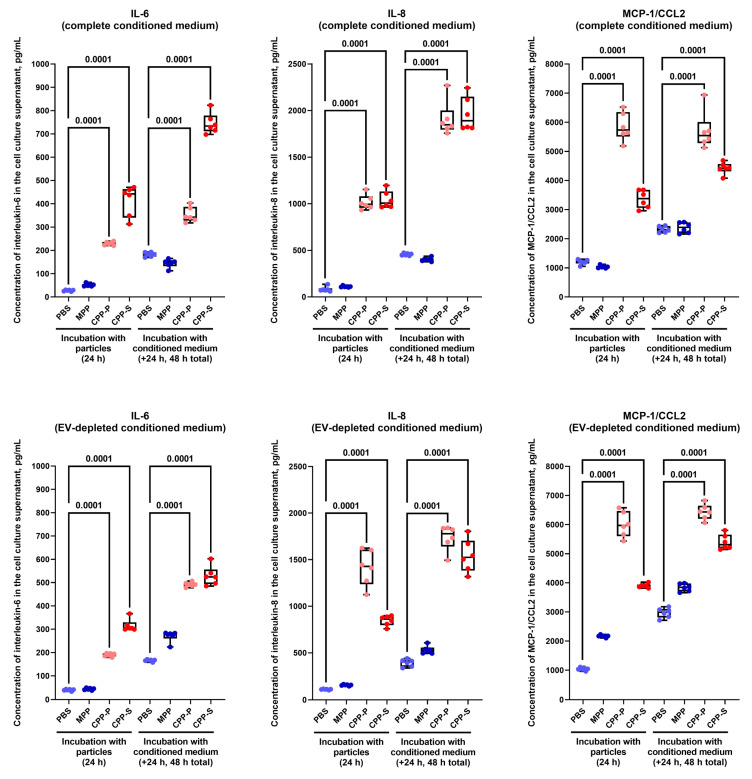
Cytokine profiling of either complete (top) or EV-depleted (bottom) conditioned medium from HCAEC incubated with MPPs, CPP-P, CPP-S, or control PBS solution for 24 h and upon another 24 h of incubation (48 h total) with intact HCAEC. Specific enzyme-linked immunosorbent assay kits for the measurement of IL-6 (left), IL-8 (centre), and MCP-1/CCL2 (right). Each plot shows the measurements after the incubation with the particles (left) and conditioned medium (right). Each dot on the plots represents one measurement (*n* = 6 measurements per group). Whiskers indicate the range, box bounds indicate the 25th–75th percentiles, and centre lines indicate the median. *p* values are provided above boxes, Kruskal–Wallis test with post hoc false discovery rate correction by the two-stage linear step-up procedure of Benjamini, Krieger, and Yekutieli.

**Figure 6 ijms-23-14941-f006:**
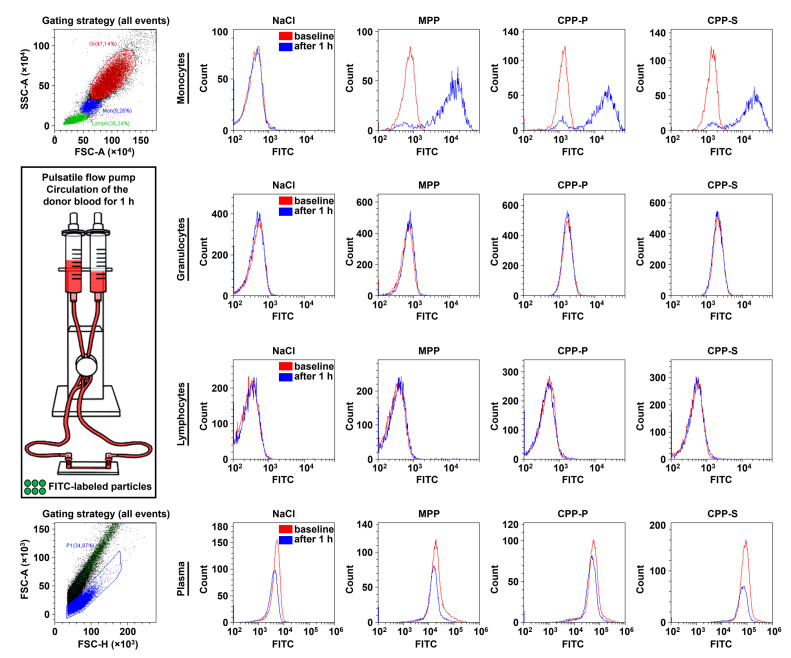
Flow cytometry analysis of CPP internalisation by means of forward scatter (FSC) and side scatter (SSC) cell clusterisation. Relative quantification of FITC-labeled MPPs, CPP-P, CPP-S, or control physiological saline (NaCl) solution has been performed immediately after their addition to the blood (baseline) and after 1 h of co-incubation of the blood with the particles under flow (15 dyn/cm^2^ shear stress). Red and blue colours are for the baseline and 1-h time points, respectively. Note that FITC-labeled particles have been well internalised by monocytes but not granulocytes or lymphocytes, concurrently with the reduction of their quantities in plasma over time.

**Figure 7 ijms-23-14941-f007:**
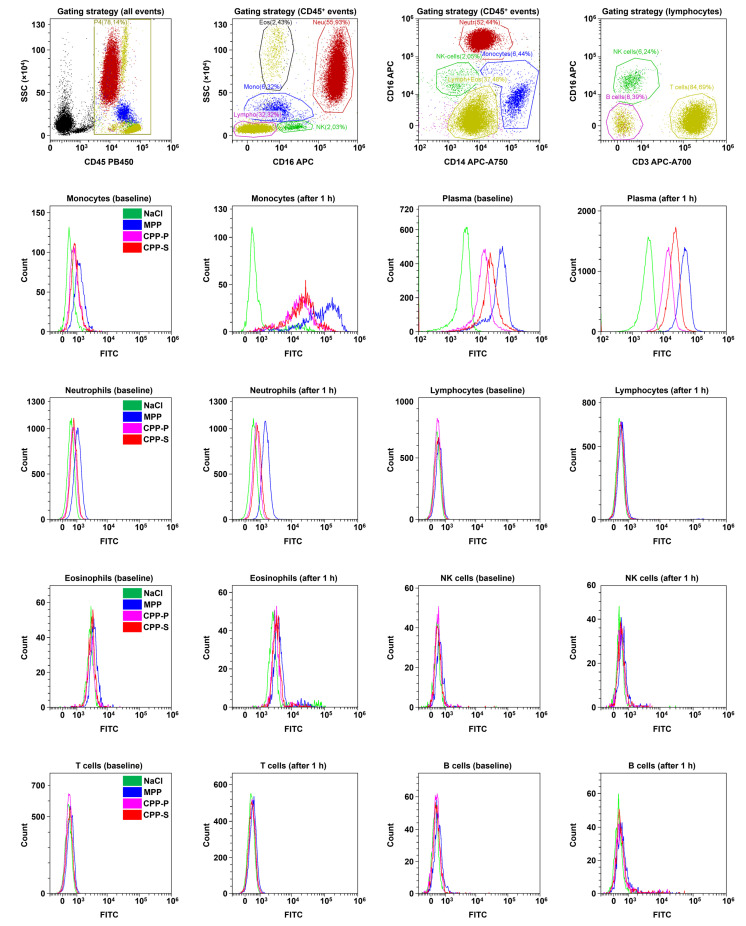
Flow cytometry analysis of CPP internalisation by means of immunophenotyping. Relative quantification of FITC-labeled MPPs, CPP-P, CPP-S, or control physiological saline (NaCl) solution have been performed immediately after their addition to the blood (baseline) and after 1 h of co-incubation of the blood with the particles under flow (15 dyn/cm^2^ shear stress). Red, blue, violet, and red colours are for NaCl, MPP, CPP-P, and CPP-S groups, respectively. As in the previous experiment, FITC-labeled particles have been well internalised by monocytes but no other leukocyte populations, and retained in the plasma after 1 h in considerable amounts.

**Figure 8 ijms-23-14941-f008:**
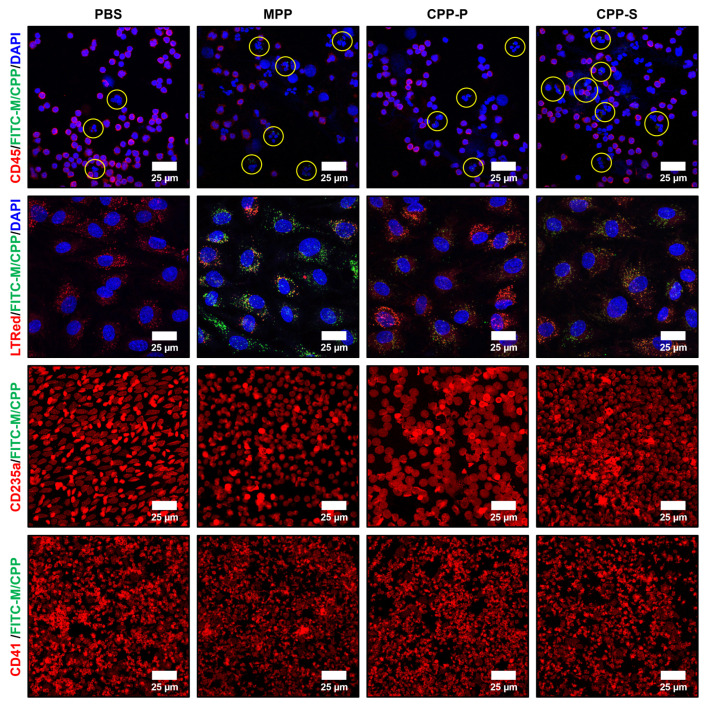
Confocal microscopy analysis of CPP internalisation after 1 h of co-incubation of the blood with FITC-labeled MPPs, CPP-P, CPP-S, or control PBS solution under flow (15 dyn/cm^2^ shear stress). Differential isolation of blood cell populations and subsequent immunostaining (CD45, CD235a, and CD41 are specific markers of leukocytes, red blood cells, and platelets, respectively, stained red). Note that the particles (stained green) are internalised by monocytes (marked by yellow circles) but not lymphocytes (distinguishable by a nuclear shape) upon the Ficoll gradient centrifugation isolation. Co-localisation of the particles (stained green) with the lysosomes (stained by LysoTracker Red) in the ECs is notable for orange or yellow colour. Magnification: ×400, scale bar: 25 µm.

**Figure 9 ijms-23-14941-f009:**
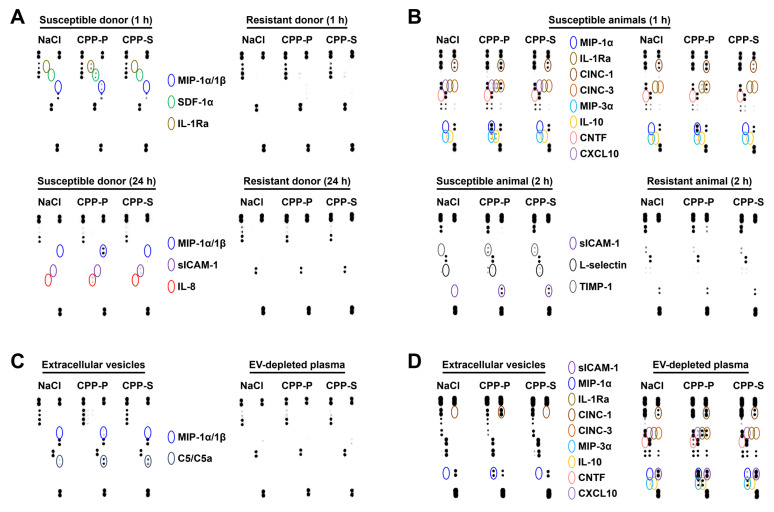
Dot blotting profiling of (**A**) plasma from the human blood co-incubated with CPP-P, CPP-S, or control physiological saline (NaCl) solution for 1 or 24 h under flow (15 dyn/cm^2^ shear stress); (**B**) plasma of rats that received intravenous injections of CPP-P, CPP-S, or control physiological saline (NaCl) solution for 1 or 2 h; (**C**) EVs and EV-depleted plasma from the human blood co-incubated with CPP-P, CPP-S, or control physiological saline (NaCl) solution for 1 h under flow (15 dyn/cm^2^ shear stress); (**D**) EVs and EV-depleted plasma from the blood of rat that received intravenous injections of CPP-P, CPP-S, or control physiological saline (NaCl) solution for 1 h. Dark blue: macrophage inflammatory protein (MIP)-1α/1β, green: stromal cell-derived factor (SDF)-1α / C-X-C motif chemokine ligand (CXCL)12, gold: interleukin-1 receptor antagonist (IL-1Ra), dark violet: soluble intercellular adhesion molecule (ICAM)-1, red: IL-8, aquamarine: complement component C5/C5a, brown: cytokine-induced neutrophil chemoattractant (CINC)-1, orange: CINC-3, light blue: MIP-3α, yellow: IL-10, pink: ciliary neurotrophic factor (CNTF), light violet: CXCL10, black: L-selectin, gray: tissue inhibitor of metalloproteinases (TIMP)-1.

**Table 1 ijms-23-14941-t001:** Bioinformatic enrichment analysis (Gene Ontology and Reactome databases) of the proteomic profiling data from HCAEC and HITAEC treated with CPP-P, CPP-S, or PBS for 24 h.

Comparison/Category (Fold Change and Number of Observed and Expected Differentially Expressed Proteins)	Upregulated				Downregulated			
	HCAEC		HITAEC		HCAEC		HITAEC	
	CPP-P vs. PBS	CPP-S vs. PBS	CPP-P vs. PBS	CPP-S vs. PBS	CPP-P vs. PBS	CPP-S vs. PBS	CPP-P vs. PBS	CPP-S vs. PBS
GO Biological Process								
Nitrogen compound metabolic process	1.49 191 vs. 129.39	1.57 188 vs. 119.51	-	-	1.55 222 vs. 143.29	1.67 230 vs. 137.73	1.47 100 vs. 67.94	1.64 105 vs. 63.92
Cellular nitrogen compound metabolic process	-	-	-	-	2.27 163 vs. 71.94	2.40 166 vs. 69.15	2.08 71 vs. 34.11	2.31 74 vs. 32.09
Cellular nitrogen compound catabolic process	-	-	-	-	2.63 19 vs. 7.23	2.59 18 vs. 6.95	-	-
Cellular nitrogen compound biosynthetic process	-	1.78 43 vs. 24.22	-	-	2.34 68 vs. 29.04	2.04 57 vs. 27.91	2.18 30 vs. 13.77	2.62 34 vs. 12.96
Regulation of nitrogen compound metabolic process	-	-	-	-	1.31 164 vs. 124.75	1.32 158 vs. 119.91	-	-
Negative regulation of nitrogen compound metabolic process	-	-	-	-	1.51 81 vs. 53.62	1.59 82 vs. 51.54	1.77 45 vs. 25.42	-
Organonitrogen compound metabolic process	1.71 167 vs. 97.39	1.78 160 vs. 89.95	-	-	1.33 143 vs. 107.85	-	-	1.62 78 vs. 48.11
Organonitrogen compound catabolic process	2.01 42 vs. 20.93	1.97 38 vs. 19.34	-	-	-	-	-	-
Organonitrogen compound biosynthetic process	2.44 65 vs. 22.43	2.68 63 vs. 23.49	-	-	2.38 67 vs. 28.16	1.99 54 vs. 27.07	2.25 30 vs. 13.35	2.39 30 vs. 12.56
Nitrogen compound transport	2.18 70 vs. 32.10	2.19 65 vs. 29.65	-	-	1.97 70 vs. 35.55	1.84 63 vs. 34.17	-	2.14 34 vs. 15.86
Cellular response to nitrogen compound	2.04 26 vs. 12.74	-	-	-	-	-	-	-
Response to nitrogen compound	2.02 44 vs. 21.79	1.99 40 vs. 20.13	-	-	-	-	-	-
Response to organonitrogen compound	2.00 40 vs. 20.02	1.95 36 vs. 18.49	-	-	-	-	-	-
Response to hydrogen peroxide	-	3.96 9 vs. 2.27	-	-	-	-	-	-
Response to reactive oxygen species	-	4.01 14 vs. 3.50	-	-	-	-	-	-
Response to oxygen-containing compound	1.69 54 vs. 31.88	1.66 49 vs. 29.45	2.08 33 vs. 15.90	-	-	-	-	-
Response to oxidative stress	2.44 19 vs. 7.77	3.62 27 vs. 7.18	-	-	-	-	-	-
Cellular response to reactive oxygen species	-	3.89 9 vs. 2.31	-	-	-	-	-	-
Cellular response to oxidative stress	-	3.56 15 vs. 4.21	-	-	-	-	-	-
Cellular respiration	4.10 15 vs. 3.66	3.84 13 vs. 3.38	-	5.40 11 vs. 2.04	-	-	-	-
Energy derivation by oxidation of organic compounds	3.01 15 vs. 4.98	3.26 15 vs. 4.60	-	4.33 12 vs. 2.77	-	-	-	-
Macroautophagy	-	-	-	-	4.55 16 vs. 3.51	-	-	-
Regulation of macroautophagy	4.53 13 vs. 2.87	4.53 12 vs. 2.65	-	-	-	-	-	-
Response to wounding	3.22 28 vs. 8.71	3.48 28 vs. 8.04	3.68 16 vs. 4.34	3.30 16 vs. 4.84	-	-	-	-
Response to stress	1.62 110 vs. 67.87	1.72 108 vs. 62.69	1.77 60 vs. 33.85	-	-	-	-	-
Response to endoplasmic reticulum stress	3.17 14 vs. 4.41	3.19 13 vs. 4.08	-	-	-	-	-	-
Response to hypoxia	3.03 18 vs. 5.94	3.46 19 vs. 5.49	-	3.63 12 vs. 3.30	-	-	-	-
Vacuolar acidification	-	-	16.42 4 vs. 0.24	-	-	-	-	-
Regulation of pH	-	-	6.43 6 vs. 0.93	-	-	-	-	-
Regulation of release of cytochrome c from mitochondria	-	-	-	9.21 5 vs. 0.54	-	-	-	-
Positive regulation of epithelial to mesenchymal transition	-	-	10.06 5 vs. 0.50	9.02 5 vs. 0.55	-	-	-	-
Regulation of peptidase activity	2.28 21 vs. 9.20	2.71 23 vs. 8.49	-	3.91 20 vs. 5.11	-	-	-	-
Negative regulation of peptidase activity	3.28 17 vs. 5.19	3.76 18 vs. 4.79	3.86 10 vs. 2.59	4.16 12 vs. 2.88	-	-	-	-
Regulation of endopeptidase activity	2.32 20 vs. 8.63	2.76 22 vs. 7.97	-	3.75 18 vs. 4.80	-	-	-	-
Negative regulation of endopeptidase activity	3.38 17 vs. 5.03	3.88 18 vs. 4.64	3.86 10 vs. 2.59	3.94 11 vs. 2.79	-	-	-	-
Regulation of hydrolase activity	1.81 38 vs. 21.02	1.91 37 vs. 19.41	-	2.99 35 vs. 11.69	-	-	-	-
Negative regulation of hydrolase activity	2.80 21 vs. 7.51	3.17 22 vs. 6.93	-	3.59 15 vs. 4.17	-	-	-	-
Positive regulation of hydrolase activity	-	-	-	2.86 19 vs. 6.65	-	-	-	-
Regulation of proteolysis	2.40 36 vs. 15.01	2.67 37 vs. 13.87	2.67 20 vs. 7.49	3.23 27 vs. 8.35	-	-	-	-
Negative regulation of proteolysis	3.01 21 vs. 6.98	3.41 22 vs. 6.45	3.45 12 vs. 3.48	3.87 15 vs. 3.88	-	-	-	-
GO Molecular Function								
Exopeptidase activity	4.73 10 vs. 2.12	5.12 10 vs. 1.95	6.63 7 vs. 1.06	-	-	-	-	-
Hydrolase activity	1.53 74 vs. 48.28	1.64 73 vs. 44.59	-	-	-	-	-	-
Metallopeptidase activity	3.31 13 vs. 3.93	3.31 12 vs. 3.63	-	-	-	-	-	-
Metalloexopeptidase activity	-	-	8.10 6 vs. 0.74	-	-	-	-	-
Calcium ion binding	2.29 34 vs. 14.85	-	2.70 20 vs. 7.41	2.91 24 vs. 8.26	-	-	-	-
Reactome pathways								
Vpr-mediated induction of apoptosis by mitochondrial outer membrane permeabilisation	49.15 3 vs. 0.06	53.22 3 vs. 0.06	-	-	-	-	-	-
Response to elevated cytosolic Ca^2+^	8.94 24 vs. 2.69	8.87 22 vs. 2.48	9.70 13 vs. 1.34	8.71 13 vs. 1.49	-	-	-	-

## Data Availability

The mass spectrometry proteomics data have been deposited to the ProteomeXchange Consortium via the PRIDE partner repository with the dataset identifier PXD038017. Other data presented in this study are available on request from the corresponding author.

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
