# Peer review of "Calciprotein Particles Cause Physiologically Significant Pro-Inflammatory Response in Endothelial Cells and Systemic Circulation"

_ijms, 2022, doi:10.3390/ijms232314941_

Round 1

Reviewer 1 Report

In the manuscript of Shishkova et al. the results of the study of the effect of  calcium protein particles (CPP) on endothelial cells are presented. The theoretical basis of the study is the concept developed by the authors, according to which CPP is an inherent buffering system in blood plasma, which prevents extra skeletal tissue calcification.  On the other hand, according to the authors, CPP plasma overload, which occurs when the kidney function is impaired, is accompanied by the absorption of these particles by endothelial cells and their damage.

As a result of the studies, reliable data were obtained on the effect of CPPs of two types - amorphous and crystalline, on the expression of genes associated with oxidative stress, nitrogen metabolism, regulation of mitochondrial functions, mitochondrial acidification, etc. The authors have demonstrated pathological paracrine effects CPPs on intact ECs. The work is characterized by a high methodological level, wide scale of research.

Comments

Introduction

In the introduction, the authors summarize the results obtained in this work, substantiating their relevance, but the conclusion about the urgent need to conduct preclinical and clinical studies of donors of Mg2+ ions concurring with Ca2+ sounds unconvincing. It is not clear what these donors are.  The statement about the particular importance of this work in relation to patients with COVID-19 should be justified.

Results

The cells were incubated with CPP-P or CPP-S during 24 hours. How stable were the CPPs, did they degrade? Did the concentration of free calcium ions in the medium change during this time? And if changed, by how much? In this regard, has the effect of an increased concentration of calcium ions in the medium on expression been studied?

What was the number of parallel  measurements in determining the effect of CPP-P or CPP-S on gene expression?

Figure 6 shows that after incubation of fluorescently stained CPP-S with plasma their number decreases. Does this mean that the particles are degrading? On the contrary, Fig. 7 shows that the content of all types of particles after an hour of incubation with plasma increases by 2-3 times. How can this be explained?

Lines 302-306. Apparently, the sequence of presentation should be changed.  

Author Response

In the manuscript of Shishkova et al. the results of the study of the effect of calcium protein particles (CPP) on endothelial cells are presented. The theoretical basis of the study is the concept developed by the authors, according to which CPP is an inherent buffering system in blood plasma, which prevents extra skeletal tissue calcification. On the other hand, according to the authors, CPP plasma overload, which occurs when the kidney function is impaired, is accompanied by the absorption of these particles by endothelial cells and their damage.

As a result of the studies, reliable data were obtained on the effect of CPPs of two types - amorphous and crystalline, on the expression of genes associated with oxidative stress, nitrogen metabolism, regulation of mitochondrial functions, mitochondrial acidification, etc. The authors have demonstrated pathological paracrine effects CPPs on intact ECs. The work is characterized by a high methodological level, wide scale of research.

Comments

Introduction

Reviewer: In the introduction, the authors summarize the results obtained in this work, substantiating their relevance, but the conclusion about the urgent need to conduct preclinical and clinical studies of donors of Mg2+ ions concurring with Ca2+ sounds unconvincing. It is not clear what these donors are.  The statement about the particular importance of this work in relation to patients with COVID-19 should be justified.

Authors: We agree with the reviewer that the last sentences of the Introduction have been over-assertive and unconvincing. We have removed the statement “This might be particularly important for the patients with COVID-19 which suffer from endothelial dysfunction, a pro-thrombotic state provoking clotting in the presence of free Ca2+ ions” (lines 85-87 in the initial manuscript) and corrected the statement “Taken together, our results underscore an urgent necessity in conducting pre-clinical and clinical trials of chemical compounds restoring the mineral buffering systems, e.g., proteinogenic amino acids replenishing depleted serum acidic proteins or donors of Mg2+ ions concurring with Ca2+ for PO43- binding” to “Taken together, our results suggest a potential clinical relevance of pre-clinical and clinical trials of chemical compounds restoring the mineral buffering systems, e.g., proteinogenic amino acids replenishing depleted serum acidic proteins or donors of Mg2+ ions concurring with Ca2+ for PO43- binding, in context of cardiovascular disease” (lines 81-85 in the revised manuscript).

Results

Reviewer: The cells were incubated with CPP-P or CPP-S during 24 hours. How stable were the CPPs, did they degrade? Did the concentration of free calcium ions in the medium change during this time? And if changed, by how much? In this regard, has the effect of an increased concentration of calcium ions in the medium on expression been studied?

Authors: We did not measure the concentration of ionised calcium (Ca2+) in the medium but the CPPs did not degrade as evident by the calcium deposition amid the cells in the static culture and internalisation of MPPs and CPPs by monocytes and ECs under flow (Figures 6-8). Further, CPPs consist of carbonate-hydroxyapatite (bioapatite formed in the living organism in presence of protein microenvironment) and MPPs consist of magnesium phosphate hydrate which both are insoluble at physiological pH (7.35 – 7.45). Therefore, we did not expect and did not detect any signs of CPP or MPP degradation over time of incubation with the cells.

However, we agree with the reviewer that comparison of the effects exerted by increased ionised calcium (Ca2+, e.g. +0.25 mmol/L above the reference range), calciprotein monomers (self-assembled ≈ 10 nm complexes of calcium ions with acidic serum proteins, i.e., mineral chaperones), and calciprotein particles (carbonate-hydroxyapatite particles formed by the binding of calcium phosphate to fetuin-A and adsorbing ambient serum proteins) represents a significant interest. This is the aim of our next study and should clearly distinguish the effects of Ca2+ ions, nanoscale clusters of Ca2+ and mineral chaperones (calciprotein monomers), and sub-microscale complexes of calcium phosphate and mineral chaperones (calciprotein particles).

Reviewer: What was the number of parallel measurements in determining the effect of CPP-P or CPP-S on gene expression?

Authors: We agree with the reviewer that we have not clearly indicated the number of measurements during the gene expression profiling. The number of biological replicates was 3 (n = 3 RNA samples collected from the cell cultures grown in different flasks, line 602 in the revised manuscript). The number of technical replicates was also 3 (n = 3 wells per PCR plate, lines 614-615 in the revised manuscript).

Reviewer: Figure 6 shows that after incubation of fluorescently stained CPP-S with plasma their number decreases. Does this mean that the particles are degrading? On the contrary, Fig. 7 shows that the content of all types of particles after an hour of incubation with plasma increases by 2-3 times. How can this be explained?

Authors: In the Figure 6, the number of CPPs in plasma reduced after 1 hour of co-incubation with the cells, probably because of their internalisation by monocytes. In the Figure 7, we noticed an opposite effect, potentially because of non-specific adsorption of different antibodies within the cocktail (as CPPs have been reported to adsorb the ambient proteins).

Reviewer: Lines 302-306. Apparently, the sequence of presentation should be changed. 

Authors: We changed “particles” to “cytokines and chemokines” (line 302 in the revised manuscript) as this was an omission from our side.

We sincerely thank the reviewer for the constructive criticism and valuable notes, which collectively helped us to improve the paper.

Reviewer 2 Report

The article describes new effects of calciprotein particles on endothelium and leucocytes. These novel data are significant for understanding the pathophisiology of disorders of calcium and phosphates metabolism, specifically in the context of chronic kidney disease, osteoporosis, and vascular diseases. The results of this study should be spread on scientific and medical community, but I have some minor suggestions for improving the article:

1. Not all the experiments have descriptions of a number of samples (n). Please, check and provide.

2. Fig. 2. I think that the deviations and statistics, at least in supplements, should be provided.

3. lane 170. I think, HITAEC also responded to stimuli, so, please revise the phrase "HITAEC did not respond to any kind of conditioned medium or EVs at the transcript level (Figure 2)"

4. Fig. 8. Oppositely to description below the Figure, in a yellow circle the CD45- cells, and in some panels without green signal, were enclosed. Why  the yellow circles were shown in a panel with PBS, the presence of particles in it were also suggested?

5. Fig. 9. "Figure 9. Dot blotting profiling of (a) human plasma co-incubated with CPP-P,". Evidently, the human blood was incubated with particles, not plasma itself. Please, revise. How many donors participated in this study?

Author Response

The article describes new effects of calciprotein particles on endothelium and leucocytes. These novel data are significant for understanding the pathophysiology of disorders of calcium and phosphates metabolism, specifically in the context of chronic kidney disease, osteoporosis, and vascular diseases. The results of this study should be spread on scientific and medical community, but I have some minor suggestions for improving the article:

Reviewer: 1. Not all the experiments have descriptions of a number of samples (n). Please, check and provide.

Authors: We agree with the reviewer that the number of replicates has not been clearly indicated in the manuscript. Here we note the number of replicates for the experiments presented in each figure:

- Figure 1 (proteomic profiling): n = 3 wells per group (line 507 in the revised manuscript). The number of technical replicates during UHPLC-MS/MS was 2 per sample (line 535 in the revised manuscript).

- Figure 2 (gene expression profiling): n = 3 cell culture flasks per group (line 602 in the revised manuscript) and 3 technical replicates (wells of the PCR plate) per each flask (lines 614-615 in the revised manuscript).

- Figure 3 (Western blotting) n = 3 cell culture flasks per group (line 604 in the revised manuscript).

- Figure 4 (dot blotting): all measurements of the cell culture supernatant are presented in the Figure 4. The dot blotting approach inherently implies 2 technical replicates (adjacent dots).

- Figure 5 (enzyme-linked immunosorbent assay): 6 measurements per group (lines 220 and 636-637 in the revised manuscript).

- Figures 6 and 7 (flow cytometry): both experiments have been performed once but experiment from the Figure 7 also confirms the results of the experiment presented in the Figure 6 (internalisation of CPPs by monocytes). The differences is that Figure 6 represents the results of forward and side scattering measurements, whilst Figure 7 shows the count of the fluorescence intensity from specific antibodies.

- Figure 8 (confocal microscopy): here we showed representative images from 4 consecutive stainings.

- Figure 9 (dot blotting): we collected the plasma from 2 donors (as shown on the image) and all cytokine measurements in human and rat plasma are represented in the figure.

Reviewer: 2. Fig. 2. I think that the deviations and statistics, at least in supplements, should be provided.

Authors: We agree with the reviewer that average ΔCt values and their standard deviations are needed for the objective analysis, and therefore added them to the revised manuscript as a Supplementary Table 3 and Supplementary Table 4 (lines 615-616 in the revised manuscript).

Reviewer: 3. Lane 170. I think, HITAEC also responded to stimuli, so, please revise the phrase "HITAEC did not respond to any kind of conditioned medium or EVs at the transcript level (Figure 2)"

Authors: We agree with the reviewer that response of intact HITAEC to the conditioned medium or extracellular vesicles from CPP-treated HITAEC was inconsistent, rather than they did not respond at all. We changed the corresponding sentence respectively (lines 168-169 in the revised manuscript). Although intact HITAEC responded to extracellular vesicle-depleted conditioned medium from CPP-S-treated HITAEC by an increased expression of certain genes related to endothelial dysfunction (VCAM1, SELE, IL6, CXCL8, MIF, SNAI2, and TWIST1), complete conditioned medium did not cause such effects and the effect was limited to CPP-S. In contrast, intact HCAEC responded to complete conditioned medium from both CPP-P- and CPP-S-treated HCAEC. Therefore, we focused on HCAEC which reacted more consistently than HITAEC to obtain the most reliable results. The number of biological replicates (cell culture flasks) and technical replicates (wells of the PCR plate) was 3 per each (lines 602 and 614-615 in the revised manuscript).

Reviewer: 4. Fig. 8. Oppositely to description below the Figure, in a yellow circle the CD45- cells, and in some panels without green signal, were enclosed. Why the yellow circles were shown in a panel with PBS, the presence of particles in it were also suggested?

Authors: In the Figure 8, monocytes are marked by a yellow circle (thus, yellow circles are presented in images from all groups). Yet, FITC-labeled MPP, CPP-P, and CPP-S emitted the green light which was strictly co-localised with monocytes and endothelial cells and was notable in all monocyte and endothelial cell images related to these groups, in particular at zoom. Monocytes were distinguished from lymphocytes by the nuclear shape, and their quantities corresponded to the expected after Ficoll isolation of peripheral blood mononuclear cells (i.e., monocytes and lymphocytes).

Reviewer: 5. Fig. 9. "Figure 9. Dot blotting profiling of (a) human plasma co-incubated with CPP-P,". Evidently, the human blood was incubated with particles, not plasma itself. Please, revise. How many donors participated in this study?

Authors: We agree with the reviewer that the legend to Figure 9 needed a correction. We replaced “human plasma co-incubated” with “plasma from the human blood co-incubated” (line 314 in the revised manuscript), as blood was co-incubated with the particles while plasma was subsequently measured by dot blotting.

We sincerely thank the reviewer for the constructive criticism and valuable notes, which collectively helped us to improve the paper.